# Deception Game: Closing the Safety-Learning Loop in Interactive Robot Autonomy

Haimin Hu[*1]   Zixu Zhang[*1]   Kensuke Nakamura[2]   Andrea Bajcsy[2]   Jaime Fernández Fisac[1]

[1]Princeton University   [2]Carnegie Mellon University

{haiminh,zixuz,jfisac}@princeton.edu,  {kensuken,abajcsy}@andrew.cmu.edu

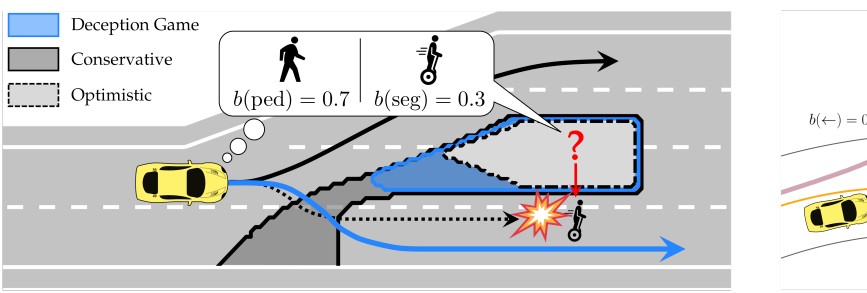

**(a)** Road crossing scenario (running example).

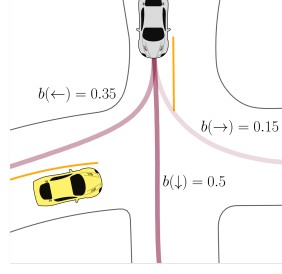

**(b)** Deep trajectory prediction.

**Figure 1:** Our proposed Deception Game framework improves safety and efficiency of uncertain, multi-modal interactions with other agents by closing the safety decision loop around the robot's runtime learning algorithm. **(a)** An autonomous vehicle is uncertain of the type of a human crossing the road. Our method computes a safe, yet non-conservative control policy by explicitly accounting for how interaction uncertainty will evolve. Color-shaded areas denote the *complement* of the safe sets projected onto the 2D plane. **(b)** Our framework can scale to systems using modern data-driven trajectory predictors such as the Motion Transformer [1].

**Abstract:** An outstanding challenge for the widespread deployment of robotic systems like autonomous vehicles is ensuring safe interaction with humans without sacrificing performance. Existing safety methods often neglect the robot's ability to learn and adapt at runtime, leading to overly conservative behavior. This paper proposes a new closed-loop paradigm for synthesizing safe control policies that explicitly account for the robot's evolving uncertainty and its ability to quickly respond to future scenarios as they arise, by jointly considering the physical dynamics and the robot's learning algorithm. We leverage adversarial reinforcement learning for tractable safety analysis under high-dimensional learning dynamics and demonstrate our framework's ability to work with both Bayesian belief propagation and implicit learning through large pre-trained neural trajectory predictors.

**Keywords:** Learning-Aware Safety Analysis, Active Information Gathering, Adversarial Reinforcement Learning.

## 1   Introduction

With the rise of autonomous vehicles on the road, robots are operating at new scales and around real people. A daily challenge for these systems is maintaining human safety in close-proximity interactions without impeding performance. This balance is made difficult by inherent *interaction uncertainty*: uncertainty induced by others' intents (e.g., does a human driver want to merge, cut behind, or stay in the lane?), responses (e.g., if the robot accelerates, how will the human react?), and semantic class (e.g., is this person a pedestrian or cyclist?). These aspects are all unknown *a priori*, so the robot must learn about them at runtime. As the robot and the human take action over time, the robot gathers more information about the human's internal state and can make more effective decisions.

---

[*]H. Hu and Z. Zhang contributed equally.

7th Conference on Robot Learning (CoRL 2023), Atlanta, USA.

Unfortunately, existing safety methods predominantly ignore the robot's ability to learn about the human during runtime interaction, instead making static modeling assumptions [2, 3, 4, 5, 6]. For example, if the robot currently believes that the human driver intends to stay in their lane, then it will compute a safety controller assuming that the human *always* intends to stay in their lane during the safety analysis horizon. Put more formally, these safety methods only account for the evolution of the physical system state while assuming that the information available to the robot remains *constant* throughout the safety analysis horizon. This simplification can lead to overly conservative robot decisions and—in extreme cases—catastrophic safety failures.

**Contributions.** We propose a novel *Deception Game* safety analysis framework that closes the loop between the robot's interactive decisions and the runtime learning process by which it updates its prediction of other agents. Safety is assessed in an augmented state space that jointly describes agents' physical motion and the robot's internal belief updates, explicitly accounting for how interaction uncertainty will evolve as a function of agents' upcoming behavior and, crucially, how rapidly the robot can detect and respond to safety-threatening tail events (Figure 1). We demonstrate the effectiveness of our approach on 200-dimensional state space problems involving black-box learning dynamics by leveraging a recently introduced approximate solution method [7] for zero-sum dynamic safety games via model-free adversarial reinforcement learning (RL).

**Related work.** A common approach to safe multi-agent interaction is to compute *robust predictions* of other agents' behavior. Forward reachability methods use (possibly learned) predictive agent models [8, 9, 10] alongside exact Hamilton-Jacobi (HJ) reachability analysis or approximations such as zonotopes [5, 11, 12, 13, 14, 15] to compute possible future states that are used as constraints by downstream motion planners [16, 17]. While safe, these methods tend to be overly conservative since they do not account for the robot's ability to take evasive maneuvers in response to the human.

Another approach is to directly synthesize *robust robot control policies*. One common tool are control barrier functions (CBFs) [18], which encode safety as an algebraic constraint in runtime optimization. However, CBFs lack general constructive methods, often requiring manual per-system design and leading to restrictive underapproximations of the safe set [19]. On the other hand, game-theoretic HJ backward reachability analysis computes the *optimal* safety value function and the corresponding safety controller for the robot [20, 21]. We ground our work in this literature, and introduce a novel formulation of game-theoretic HJ reachability in belief space.

To date, all provably safe human-robot interaction schemes treat the human model as *static* during the entire safety analysis horizon. In other words, they assume that the robot will never gain new information about the human's behavior; we refer to this as *open-loop* safety in the *information space*. In reality, however, robots can gain information about other agents intent', semantic class, etc., during interaction. By reasoning about the *future information* they might receive, robots can afford to be less conservative: for example, in occluded environments, safety can be maintained less conservatively by leveraging the future observability of hypothetical yet-undetected objects [22, 23]. Our approach accounts for runtime inference, yielding a novel *closed-loop-information* safety analysis.

Finally, any approach that considers multiple agents and/or hypotheses will face challenges with the curse of dimensionality. Although HJ reachability has traditionally been intractable for systems with more than five continuous state variables, neural approximations have shown promise in scaling to high-dimensional systems [24, 25, 7]. In this work, we leverage recent advancements in adversarial reinforcement learning approximations [7] to scale our novel safety problem.

## 2   Preliminaries and Problem Formulation

Let the discrete-time nonlinear dynamics of an "ego" autonomous robot ($e$) interacting with an "opponent" agent ($o$) be $x_{t+1} = f(x_t, u_t^e, u_t^o, w_t)$ where $x_t = (x_t^e, x_t^o) \in \mathcal{X} \subseteq \mathbb{R}^n$ is the joint state vector. The control vectors of the robot and the opponent are $u_t^e \in \mathcal{U}^e \subset \mathbb{R}^{m_e}$ and $u_t^o \in \mathcal{U}^o \subset \mathbb{R}^{m_o}$. Finally, $w_t \in \mathcal{W} \subset \mathbb{R}^{m_w}$ is a bounded process noise (external disturbance), and $f : \mathcal{X} \times \mathcal{U}^e \times \mathcal{U}^o \times \mathcal{W} \to \mathcal{X}$ describes the physical system dynamics.

**Background: Hamilton-Jacobi reachability.** Game-theoretic HJ reachability analysis [20, 26] uses robust optimal control theory to compute provable policies to reach and/or avoid regions of the state space. Define the target and failure sets as $\mathcal{T} := \{x \mid \ell(x) \geq 0\} \subseteq \mathbb{R}^n$ and $\mathcal{F} := \{x \mid g(x) < 0\} \subseteq \mathbb{R}^n$, encoded by Lipschitz-continuous *margin functions* $\ell(\cdot)$ and $g(\cdot)$. HJ analysis captures the opponent's worst-case behavior by formulating an infinite-horizon zero-sum dynamic game. In discrete time, the game's solution can be obtained via the fixed-point Isaacs equation [27] (the game-theoretic counterpart to the Bellman equation):

$$V(x) = \min \left\{ g(x), \max \left\{ \ell(x), \max_{u^e \in \mathcal{U}^e} \min_{u^o \in \mathcal{U}^o, w \in \mathcal{W}} V\big(f(x, u^e, u^o, w)\big) \right\} \right\}. \tag{1}$$

The value function $V(\cdot)$ encodes the reach-avoid set $\mathcal{RA}(\mathcal{T}, \mathcal{F}) := \{x \mid V(x) \geq 0\}$, from which the ego agent is guaranteed a policy to safely reach the target set without entering the failure set. Given a value function $V(\cdot)$, its zero superlevel set contains the ego's recursively safe states, and the optimal ego and opponent policies are obtained by taking argmax/argmin controls in (1) [20].

**Opponent uncertainty.** We model opponent agents as taking actions from an *uncertain control set* affected by their unknown individual characteristics $\theta \in \Theta$. We refer to $\theta$ as the opponent's *type*, following game theory literature [28]. This type $\theta$ may encode aspects about the agent, including its *semantic class* (e.g., a pedestrian or a cyclist) and its *internal state* (e.g., intended destination, attention state). We assume that the hypothesis space $\Theta$ is discrete, and that each hypothesized type $\theta$ is associated with a known (possibly state-dependent) control set $\mathcal{U}_\theta^o(x_t)$ of *admissible* actions for an opponent agent of this type.

**Reducing uncertainty via learning.** The robot represents its uncertainty over the agent type $\theta \in \Theta$ through a probabilistic *belief* $b(\theta) \in \Delta$. Since $\Theta$ is a finite set, $b$ is a *categorical distribution* and the belief space $\Delta$ is a $(|\Theta| - 1)$-simplex. The belief evolves over time as the ego agent collects new observations (e.g., observing the opponent nudging into its lane). Our approach is agnostic to the specific runtime learning algorithm used for propagating the belief state, which includes, among others, Bayesian inference and transformer neural networks [1]. Thus, we refer to the abstracted evolution of the belief state as the *learning dynamics* of the form

$$b_{t+1} = f_L(b_t, y_t), \qquad y_t = h(x_t, u_t^o, v_t), \tag{2}$$

where $y_t \in \mathcal{Y} \subseteq \mathbb{R}^{n_y}$ is the robot's indirect observation of the opponent at time $t$ from state $x_t$, $v_t \in \mathcal{V} \subset \mathbb{R}^{m_v}$ is a bounded measurement noise, and $h : \mathcal{X} \times \mathcal{U}^o \times \mathcal{V} \to \mathcal{Y}$ is the observation model. Defining the joint state vector $z := (x, b)$, we can combine the physical dynamics function $f$ and the learning dynamics $f_L$ from (2) into a joint dynamical system:

$$z_{t+1} = F(z_t, u_t^e, u_t^o, w_t, v_t) := \begin{bmatrix} f(x_t, u_t^e, u_t^o, w_t) \\ f_L\big(b_t, h(x_t, u_t^o, v_t)\big) \end{bmatrix}, \tag{3}$$

which compounds the physical evolution of the system with the robot's belief update. We note that the learning dynamics need not impose (or be aware of) the type-specific control bounds $\mathcal{U}_\theta^o(x_t)$ used for safety analysis purposes.

*Running example:* In Figure 1, the autonomous vehicle (ego) is uncertain about whether the crossing human (opponent) is a pedestrian or a Segway rider. We model the opponent's unknown type as $\theta \in \Theta := \{\text{ped}, \text{seg}\}$, with associated control sets $\mathcal{U}_{\text{ped}}^o = [-0.75, 0.75] \times [-2, 2]$ m/s and $\mathcal{U}_{\text{seg}}^o = [-0.75, 0.75] \times [-8, 0]$ m/s. The joint state space is defined as $z := (x^e, x^o, b(\theta = \text{ped})) \in \mathbb{R}^8$ with $x^e \in \mathbb{R}^5$ and $x^o \in \mathbb{R}^2$. The robot uses Bayesian inference as the learning dynamics $f_L$.

## 3 Closing the Safety Loop Around Runtime Learning

Our approach reduces conservativeness by explicitly modeling the *dynamic coupling* between the opponent's control bound and the robot's internal runtime learning. For example, imagine that the robot begins with a strong prior that humans do *not* cross highways: by the time the human *does* begin to cross the highway, the robot must have observed enough evidence that suggests that

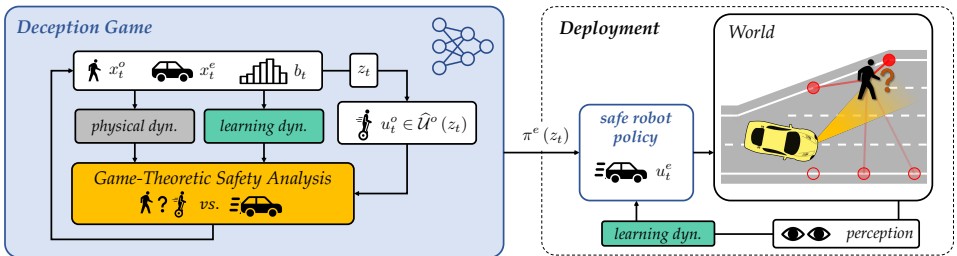

**Figure 2:** Deployment of the Deception Game policy trained with adversarial deep RL in closed-loop with a pretrained deep neural network learning dynamics model. The robot policy takes as inputs both the physical state $x_t$ and belief state $b_t$, and operates under the *inference hypothesis* that the opponent behaves according to the inferred control bound $\widehat{\mathcal{U}}^o(z_t)$ imposed by the learning dynamics.

the human's true intent *might* be to cross. Note that this results in a belief-based restriction of the opponent's behavior that is not *causal*, but *inferential*: it is based on the premise that an agent cannot take actions only explained by $\theta$ without the robot's belief already placing nonnegligible probability on this type. We formalize this below as the *inference hypothesis*.

**Assumption 1** (Inference Hypothesis). *At each time $t$, the robot's belief $b_t$ must have assigned no less than $\epsilon_\theta$ probability to at least one type $\theta$ consistent with the opponent's next action $u_t^o$. That is, the opponent's action $u_t^o$ from joint state $z_t$ must belong to the* inferred control bound *given by*

$$\widehat{\mathcal{U}}^o(z_t) := \bigcup_{\theta \in \Theta} \widehat{\mathcal{U}}_\theta^o(z_t), \quad \widehat{\mathcal{U}}_\theta^o(z_t) := \begin{cases} \mathcal{U}_\theta^o(x_t), & \text{if } b_t(\theta) \geq \epsilon_\theta \\ \emptyset, & \text{if } b_t(\theta) < \epsilon_\theta \end{cases} \tag{4}$$

*where $\epsilon_\theta \geq 0$ is a designer-specified threshold capturing the reliability of the runtime inference.*

Assumption 1 implies that when $b_t(\theta)$ falls below $\epsilon_\theta$ the corresponding control set $\mathcal{U}_\theta^o(x_t)$ can be *temporarily omitted*, reducing the opponent action uncertainty to a smaller inferred bound $\widehat{\mathcal{U}}^o(z_t)$. The belief threshold $\epsilon_\theta$ encodes the designers' confidence in the reliability of the robot's runtime inference. A suitable value can be determined empirically by evaluating the inference mechanism on a large dataset, or it may be provided by an upstream component in the autonomy stack as part of a "contract" [29]. As $\epsilon_\theta$ approaches zero (no confidence in the robot's inference), we recover the purely robust formulation, where the robot safeguards against all hypotheses regardless of its belief.

**Remark 1.** *Unlike scenario pruning approaches [30, 31, 32], within our framework no hypothesis is ever permanently ruled out, but only provisionally set aside until and unless its probability increases. Specifically, a previously small $b(\theta)$ may regain probability and surpass threshold $\epsilon_\theta$, at which point the corresponding control set $\mathcal{U}_\theta^o(x_t)$ is immediately added back to the overall inferred control bound $\widehat{\mathcal{U}}^o(z_t)$. This possibility is factored into our safety analysis via the learning dynamics.*

**Belief-space HJ reachability: formalizing learning-aware safety.** The coupling between the robot's learning dynamics and the opponent's action uncertainty induces a new safety problem: a *belief-space reach-avoid game* in which the ego agent attempts to robustly ensure safety against the worst-case behavior of an opponent whose actions must be plausible in light of past behavior, yet may be strategically deceptive. To model this game, we extend the game-theoretic HJ reachability formulation (Section 2) to the joint physical–belief state $z$. Although the target set $\mathcal{T} := \{z \mid \ell(x) \geq 0\} \subseteq \mathcal{X} \times \Delta$ and failure set $\mathcal{F} := \{z \mid g(x) < 0\} \subseteq \mathcal{X} \times \Delta$ are still defined in terms of only the *physical state*, the reach-avoid game *entangles* physical and belief states through the joint dynamics (3). This interdependence is characterized by a new *belief-space Isaacs equation*:

$$V(z) = \min \left\{ g(z), \max \left\{ \ell(z), \max_{u^e \in \mathcal{U}^e} \min_{u^o \in \widehat{\mathcal{U}}^o(z), w \in \mathcal{W}, v \in \mathcal{V}} V\left(F(z, u^e, u^o, w, v)\right) \right\} \right\}. \tag{5}$$

The key differences with respect to the standard Isaacs equation (1) are highlighted in red. In particular, the state space now is defined over $z$ (the joint state that additionally includes the robot's belief $b$) and the inferred control bound $\widehat{\mathcal{U}}^o(z)$ becomes a *function* of the joint state instead of fixed

*a priori*. We also are robust to *observation noise* as modelled by $v$. Note that the ego and opponent controls given by solving (5) are optimal in the sense of a zero-sum partially observable stochastic game [33] (or dual control [34, 35]): they not only aim to influence the physical state $x$, but also the ego's uncertainty about the opponent, encoded by the belief state $b$. The ego's control $u^e$ strategically forces the opponent to reveal information about its type $\theta$ whenever reducing the uncertainty is critical to safely reaching the target. The opponent's control $u^o$ is also dual: to sabotage the ego's reach-avoid task completion, it may behave deceptively to maintain strategic ambiguity that it can exploit later, for example, by forcing a collision (safety failure). This is formalized in the result below, which is a direct consequence of HJ reachability theory [21].

**Proposition 1.** *If the opponent satisfies the inferred control bound* (4), *i.e* $u_t^o \in \widehat{\mathcal{U}}^o(z_t), \forall t \geq 0$, *then the superlevel set* $\{z \mid V(z) \geq 0\}$ *is a robust reach-avoid set from which the robot has a guaranteed control strategy—the maximizer of* (5)—*to drive the joint system* (3) *into* $\mathcal{T}$ *while avoiding* $\mathcal{F}$.

**Remark 2.** *The above proposition guarantees that, as long as the inference hypothesis encoded through* (4) *holds, the reach-avoid policy is* recursively feasible *when joint state $z$ is initialized within the reach-avoid set* $\{z \mid V(z) \geq 0\}$. *This naturally accounts for* future *belief updates, which may discard low-probability hypotheses or add one back as it regains probability (see Remark 1).*

*Running example:* The ego vehicle's target set is defined as $\mathcal{T} = \{z \mid p_x^e \geq 85\,\mathrm{m}\}$, where $p_x^e$ denotes the ego vehicle's longitudinal position. Failure set $\mathcal{F}$ captures the ego colliding with the opponent or driving out of the road boundary. The *complement* of the belief-space reach-avoid set $\mathcal{RA}(\mathcal{T}, \mathcal{F})$ computed numerically (with simplified motion models) on a grid is visualized in blue in Figure 1.

**Synthesizing safe robot policies.** Given a pre-computed value function $V(z)$, and any joint physical-belief state $z$, the optimal agent policies are obtained by argmin/argmax over (5). The resultant value functions and safety policies can seamlessly integrate with existing safety-aware control frameworks (e.g., HJ value functions are optimal Control Barrier Functions (CBFs) [18, 36], and the HJ safety policy can be used as a fallback controller [37, 38, 39]).

**Scaling-up: embracing model-free deep RL.** Since the Isaacs equation (5) is generally intractable to solve exactly, and an explicit Markovian model of the belief dynamics may not always be available, we introduce an adversarial RL approach to approximate a time-discount variant of (5) based on the recently developed Iterative Soft Adversarial Actor-Critic for Safety (ISAACS) framework [7]. During training, a runtime learning algorithm is queried at each time step with the current observation ($y$ or, depending on the algorithm, a *history* of recent observations) to produce an updated belief state $b_{t+1}$, without the explicit requirement to be Markovian as in (2). The physics simulator enforces the *inference hypothesis* during training by projecting the opponent's action $\pi^o(z)$ onto the inferred control bound $\widetilde{\mathcal{U}}^o(z)$ defined in (4). At runtime, the ego agent receives the current state estimate $x$ and an observation $y$ of the opponent, queries the same learning algorithm used in training for a belief state $b$, and executes control action $u^e = \pi^e(z)$. Figure 2 presents a schematic of our proposed safe control framework.

## 4 Case Studies

In this section, we illustrate the applicability and scalability of our approach in three simulated driving examples that differ in problem scale, computation approach, and runtime learning mechanism. *Our main hypothesis is that the proposed Deception Game policy yields a planning performance close to an optimistic baseline while achieving a much lower failure rate.*

### 4.1 Low-dim (5D): Exact solution with Bayesian learning dynamics

We first consider a simplified variant of the running example. The ego vehicle's motion is described by a 3D point mass model and the human is restricted to travel along the $p_y^o = 90\,\mathrm{m}$ line, with possible agent classes $\theta \in \{\mathrm{ped}, \mathrm{seg}\}$. The robot learns via updating a Bayesian belief: $b_{t+1}(\theta) \propto P(u_t^o \mid x_t; \theta) b_t(\theta)$. The likelihood function is a Gaussian $P(u_t^o \mid x_t; \theta) = \mathcal{N}(\mu_\theta, \Sigma)$

| Metric/Method | Opponent policy | MAP (optimistic) | Contingency | Robust (w/o learning) | Deception Game (ours) |
|---|---|---|---|---|---|
| Failure rate | Modeled ($\epsilon_\theta = 0.2$) | 11 % | 3.1 % | **0 %** | **0 %** |
| Completion time (s) | | **4.49 $\pm$ 0.6** | 5.2 $\pm$ 1.12 | 6.27 $\pm$ 0.86 | 4.73 $\pm$ 0.97 |
| Failure rate | Unmodeled ($\tilde{\epsilon}_\theta = 0.05$) | 16.4 % | 3.7 % | **0%** | 3.3 % |
| Completion time (s) | | **4.54 $\pm$ 0.6** | 5.13 $\pm$ 1.14 | 6.26 $\pm$ 0.85 | 4.81 $\pm$ 0.85 |

**Table 1:** Case study: Low-dim (5D). Failure rate and mean completion time in 1000 randomized trials with policies synthesized with numerical dynamic programming [40]. The Deception Game is computed with $\epsilon_\theta = 0.2$. We also tested with an unmodeled human that violated the inference hypothesis by taking actions from types as unlikely as $\tilde{\epsilon}_\theta = 0.05$. In the modeled human case, the proposed Deception Game policy ensures safety in all trials while closely matching the performance of an optimistic, but unsafe MAP policy. In the unmodeled case, the Deception Game policy incurs a non-zero failure rate, although lower than the MAP and contingency baselines.

where $\mu_\theta$ is the average control of the bound $\mathcal{U}_\theta^o(x_t)$. The joint system (3) is 5-dimensional, which lends itself to numerical grid-based dynamic programming [40] for solving Isaacs equation (5) to high accuracy. We compare our Deception Game policy to three baselines (all synthesized with the same grid resolution as the proposed method): (1) a **maximum a posteriori (MAP)** policy [41], which *optimistically* uses a standard (belief-less) reach-avoid policy based on the MAP estimate $\hat{\theta}_t \in \arg\max_{\theta \in \Theta} b_t(\theta)$ during each planning cycle, (2) a **contingency** policy [30, 31], which discards *currently* unlikely hypotheses for which $b(\theta) < \epsilon_\theta$ while safeguarding against all remaining hypotheses, and (3) a **robust** policy that *conservatively* safeguards against all hypotheses $\theta \in \Theta$ at all times. Note that the Robust baseline is equivalent to the *optimal* control barrier function [18]. We consider two metrics: the *failure rate*, defined as $N_{\text{fail}}/N_{\text{trial}} \times 100\%$, where $N_{\text{trial}}$ is the number of trials and $N_{\text{fail}}$ is the number of failed trials in which $z_t \in \mathcal{F}$ for some $t$, and the *completion time* averaged across all trials, which measures a policy's ability to make progress. A trial is considered complete if $p_x^e \geq 100$ m. For each policy, we simulated under the same random seed $N_{\text{trial}} = 1000$ randomized scenarios, each with a different initial state, prior belief, and the (hidden) opponent type. Table 1 displays results obtained from all 1000 trials. The Deception Game policy maintained a zero failure rate when the inference hypothesis held, which empirically validated Proposition 1, and achieved an average completion time close to that of the optimistic but unsafe MAP baseline.

## 4.2 High-dim (18D): Approximate solution with Bayesian learning dynamics

Now, we turn to the full setup of the running example with the 7D physical system (the ego and opponent are modeled as a 5D kinematic bicycle model and a 2D particle with velocity control) and two agent class hypotheses $\theta \in \{\text{ped}, \text{seg}\}$, and *additionally* model the human's intent with hidden goal position $\theta_{\text{goal}} \in \Theta_{\text{goal}} := \{g_i\}_{i=1}^{N_{\text{goal}}}$, respectively, where $g_i \in \mathbb{R}^2$ denote each of $N_{\text{goal}} = 9$ goal hypotheses scattered along the road boundary. Since the resulting 18-dimensional state space is intractable for exact dynamic programming, we apply the deep RL framework in Sec. 3. We simulated 1000 randomized scenarios with different initial states, prior belief, opponent types, and latent goals for each policy under the same random seed. The simulation was repeated for four different human policies: (1). **Belief-space adversarial (modeled):** The human uses adversarial policy $\pi^o(z)$ trained with the belief-space deep RL that approximates (5) and was projected to the inferred control bound $\widehat{\mathcal{U}}^o(z)$ at simulation time, (2). **Belief-space adversarial (unmodeled):** Same as (1) but without the projection step, thereby allowing violations of the inference hypothesis, (3). **Non-belief adversarial:** The opponent uses adversarial policy $\pi^o(x)$ trained with the non-belief deep RL that approximates (1) while the ego safeguards against all hypotheses, and (4). **Noisily-rational:** The human uses a perturbed LQR policy to reach a goal position (randomized across trials). The failure rate and completion time results are shown in Table 2 and Figure 3, respectively. The proposed Deception Game policy outperforms both the MAP and contingency baselines in all scenarios in terms of the failure rate. Remarkably, the completion time statistics of the Deception Game policy closely resemble that of the optimistic MAP policy in all four scenarios. The robust policy, albeit safe, was, however, overly conservative and unable to complete the task within 10 seconds in most trials. These results empirically support our main hypothesis.

In Figure 4, we examine one representative simulation trial of the running example initialized with a uniform prior belief. As the ego vehicle approached the human during $t \in [0, 4.4]$ s, the human

| Metric/Method | Opponent policy | MAP (optimistic) | Contingency | Robust (w/o learning) | Deception Game (ours) |
|---|---|---|---|---|---|
| Failure rate (Incompletion rate) | Belief Adv. (Modeled) | 40.7 % (40.7 %) | 31.2 % (39.5 %) | 22.3 % (84.6 %) | **5.7 % (5.9 %)** |
| | Belief Adv. (Unmodeled) | 62.5 % (62.5 %) | 51.3 % (53.6 %) | **8.1 % (84.4 %)** | 36.4 % (**36.4 %**) |
| | Non-belief Adv. | 5.2 % (5.2 %) | 1.5 % (2.3 %) | 7.4 % (81.2 %) | **1 % (1.5 %)** |
| | Noisily-rational | 10.3 % (10.3 %) | 6 % (14.5 %) | 5.2 % (54.5 %) | **0.5 % (0.7 %)** |

**Table 2:** Case study: Running example (18D). Failure rate and task incompletion rate (counting *both* failures and time-outs) in 1000 randomized trials with deep RL–trained policies. The Deception Game policy achieves a consistently lower failure rate than the MAP and contingency baselines and leads completion across the board.

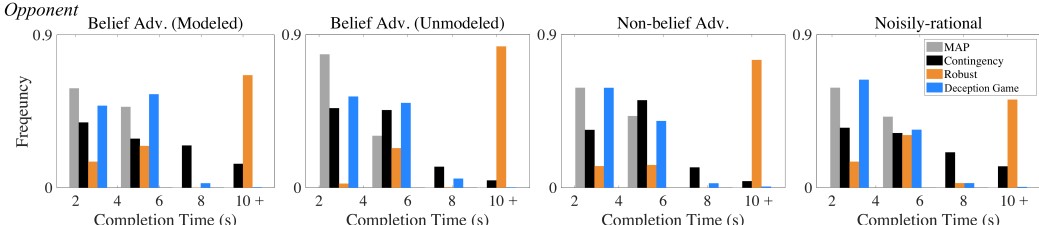

**Figure 3:** Case study: Running example (18D). Completion time over 1000 randomized trials. Approximations via deep adversarial RL. Deception Game policy performance closely matches the optimistic MAP baseline. The robust baseline is overly conservative and often times out ( $> 10$ s).

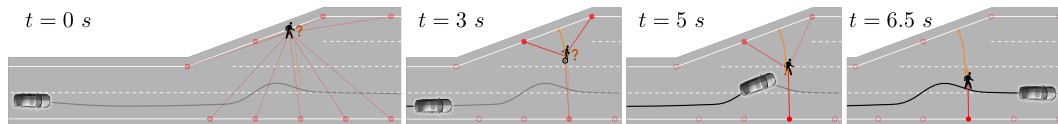

**Figure 4:** Case study: Running example (18D). Simulation snapshots and $b(\text{type})$ over time. Both agents use the optimal Deception Game policies synthesized with adversarial RL. Red circles denote the human's goal hypotheses and color intensity represents their belief probability. A line between the human and a goal $g_i$ means that $b(g_i) \geq \epsilon_\theta$. The pedestrian and Segway icons indicate the ground-truth human type. The orange question mark implies that neither hypothesis has a probability below the threshold. Despite the human's adversarial and deceptive motion, the ego vehicle managed to safely complete the task.

*deceptively* alternated between applying pedestrian and Segway actions in order to hide its type. The ego vehicle then deliberately turned left, causing the adversarial human to move upwards and be exposed as a pedestrian. Due to the pedestrian's limited ability to move downwards compared to the Segway type, the ego vehicle could safely turn right and pass in front of the human.

### 4.3 High-dim (200D): Implicit learning dynamics from neural predictor

Finally, we demonstrate that our method can scale to a data-driven, non-Markovian learning process. Consider a scenario from the Waymo Open Motion Dataset (WOMD) [42], where an autonomous car aims to cross the intersection without colliding with the oncoming human-driven car or violating the road restrictions (left column, Fig. 5). The robot's learning process is governed by a *pretrained* Motion Transformer (MTR) [1] trajectory predictor, which at each timestep produces a Gaussian mixture model of 64 state trajectories given the scene's history. Using a proportional controller to invert the dynamics, the robot infers the human's action (acceleration, steering angle) from these state predictions. We use the inferred actions as the mean of each mixture component (mode $\theta$). Following (4), we add a small $d_t^o(\theta)$ around each predicted action when constructing the predictive control bound $\widehat{\mathcal{U}}_\theta^o$. Since the robot will repeatedly invoke the MTR to forecast the human's behavior, and these predictions will inevitably change over time during the closed-loop interaction, this defines the robot's *implicit* learning dynamics $f_L(\cdot)$. Ultimately, this problem formulation has a 200-dimensional joint state space: an 8D physical state and a belief vector comprised of 64 mixture components, each with a 2D mean and a scalar weight, totaling $64 \times 3 = 192$ belief dimensions.

Both the robot and the human's control policies were trained using the deep RL approach in section 3, taking as inputs the physical states, predicted nominal actions, and the corresponding probabilities. We implement two baselines: (1) a **robust** policy that safeguards against the entire human action space $\mathcal{U}^o$, and (2) an **iterative linear quadratic regulator (ILQR)** policy that re-plans using

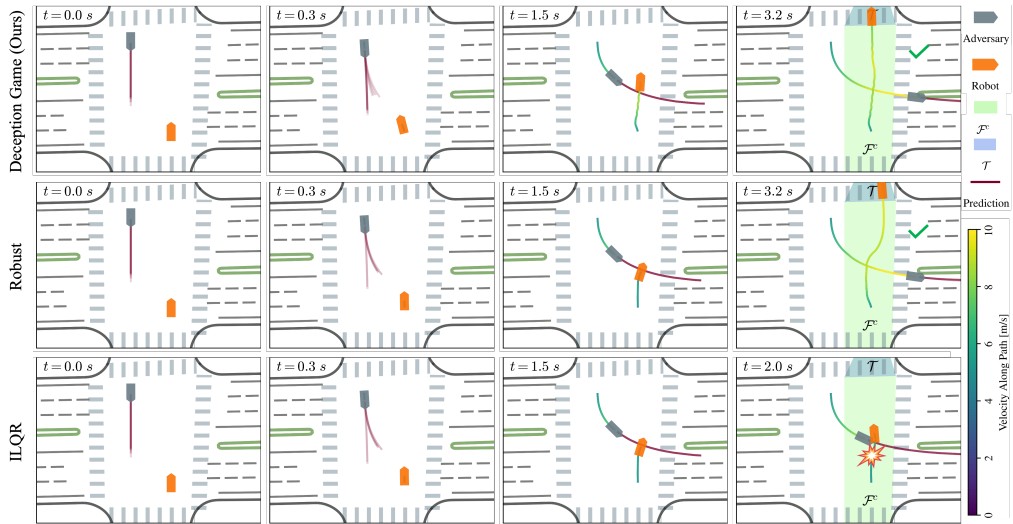

**Figure 5:** Case study: neural predictor (200D). The adversary suddenly turned left in the intersection. **Deception Game** (top) and **Robust** policy (middle) safely reached the target $\mathcal{T}$, while **ILQR** (bottom) leads to a collision. Compared to the robust policy, our proposed approach is also smoother.

MTR predictions at each timestep. By closing the safety and learning loop, the Deception Game policy produces proactive ego behaviors, preparing the robot for potential shifts in belief. For example, in Figure 5, at $t = 0.3$ s, the robot predicted the human's left turn and maintained a lower velocity for a larger safety margin. This allowed the robot to safely pass the human later at $t = 1.5$ s when it slowed down, attempting to block the robot. In contrast, the robust baseline produced more radical evasive actions to protect against the adversary regardless of belief. In Appendix C, we include additional preliminary results to look closer at our Deception Game policy against two baselines under different initial states. Moreover, we also demonstrate its generalizability to non-adversarial human behaviors by replaying ground-truth trajectories from WOMD, as well as its scalability to multiple-agent scenarios under the commonly used pairwise decomposition assumption [43, 44].

## 5 Limitations and Future Work

While our empirical results indicate that our Deception Game improves robot efficiency and maintains a low failure rate compared to baselines, we cannot provide theoretical safety guarantees when the solution is approximated via RL. In addition, our approach relies on a designer-specified threshold $\epsilon_\theta$ for the control bound, which requires careful construction. Moreover, our MTR case study only evaluates one intersection scenario with hand-crafted target and failure sets. One direction for future work is to leverage the learned latent information (about the scene or interactions) from the scene-centric trajectory prediction models [45, 46, 47, 48] to improve generalization. We are also interested in incorporating our approach with a full autonomy stack, accounting for raw sensor data (e.g. camera images) and imperfect state observations (e.g. occlusions [22]). Finally, we see an open opportunity to extend our methodology for other pressing issues beyond robot safety, e.g., detecting and preventing manipulative behaviors in generative AI (such as large language models).

## 6 Conclusion

In this paper, we present the Deception Game, a novel safety framework that closes the loop between the robot's prediction-planning-control pipeline and its runtime learning process by *jointly* reasoning over agents' physical states and the robot's belief states. Our approach builds upon a game theoretic formulation and can scale up to high-dimensional problems and handle implicit learning dynamics via model-free adversarial reinforcement learning. Our experiment results indicate that the proposed method works with different types of interaction uncertainty, can be well combined with off-the-shelf prediction models, and plan efficient and safe trajectories.

**Acknowledgments**

The authors thank Hongkun Liu from the University of Science of Technology Beijing for providing insightful suggestions on trajectory prediction and behavior forecasting.

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

# A    Comparison of Learning-Aware Safety Methods

This section provides a table comparing our proposed Deception Game framework with recent learning-aware safe planning methods. To the best of our knowledge, our method is the first safety analysis framework that jointly reasons the agent's physical states and the robot's belief states in a closed-loop fashion and can scale up to high-dimensional systems with implicit learning dynamics.

**Table 3:** Comparison of learning-aware safe planning methods.

| Feature/Method | Peters et al. [32] | Tian et al. [2] | Hu et al. [49] | Bajcsy et al. [30] | Zhang et al. [22] | Packer et al. [23] | **Ours** |
|---|---|---|---|---|---|---|---|
| Recursive safety guarantees | N | N | **Y** | **Y** | **Y** | N | **Y** |
| Active information gathering | N | N | **Y** | N | **Y** | **Y** | **Y** |
| Uncertainty-dependent safety analysis | N/A | N | N/A | **Y** | **Y** | N/A | **Y** |
| Belief refinement based on observations | N | **Y** | **Y** | **Y** | N/A | N/A | **Y** |
| Scaling to high dimension ($n_z > 10$) | **Y** | N | N | N | N | **Y** | **Y** |
| Allowing implicit learning dynamics | N | N | N | **Y** | N | **Y** | **Y** |
| Allowing continuous hypotheses space | N | N | **Y** | N | N | N/A | N |
| Fully online policy computation | **Y** | N | **Y** | N | **Y** | N | N |

# B    When the Prior is Biased and the Inference Hypothesis is Violated

We provide another example of the high-dim (18D) scenario, where the robot's prior belief is initially biased towards believing that the human will *not* cross the road, and that the human is allowed to take actions that *violate the inference hypothesis (Assumption 1)*. To this end, we give the robot a biased prior belief on the human's intent, in which the goal encoding the human not crossing the road ($g_8$) receives most of the probability mass (0.92), and the prior probabilities of the remaining goals are set to 0.01, which is below threshold $\epsilon_\theta = 0.05$. The simulation snapshots are plotted in Figure 6. We observe that as the robot receives new observations of the human's states and actions, its runtime learning algorithm gradually adjusts the belief to better explain the observations. As a result, goal $g_3$ on the other side of the road starts gaining probability, and its associated control bound is subsequently added to the overall inferred control bound. At $t = 6$ s, all hypothesized goals in the upper part of the road are temporarily thresholded out as the human moves down. Then, the human abruptly begins to move upwards toward $g_7$, violating the inference hypothesis (indicated by the yellow exclamation mark). In light of this highly unexpected behavior, the robot's runtime inference quickly catches on, which reactivates hypotheses $g_7$ and $g_8$ in the robot's belief. The Deception Game policy then generates cautious robot motion based on the updated belief. In spite of the unfavorable conditions of a biased prior and abrupt, assumption-breaking human behavior, the robot avoids colliding with the human and safely reaches its target. This case study indicates that the Deception Game policy offers some practical robustness to inaccurate prior beliefs as well as abrupt changes in agent behavior inconsistent with the inference hypothesis.

We also simulate trajectories under the same initial condition and human policy when the ego vehicle uses the three baseline policies. Figure 7 displays the trajectories with the MAP policy. Due to the overly optimistic assumption that only the most likely hypothesis is considered during decision-making, the ego vehicle does not take precautions until the last instant, when the runtime learning indicates that the human is moving downwards, by which time it is already too late to avoid colliding with the adversarial human. Simulation snapshots when the ego uses the contingency baseline policy are shown in Figure 8, which also constitutes a failure case. The main reason for the loss of safety, in this case, is due to the biased prior—the algorithm is initialized with $g_8$ being the only active goal hypothesis, with all other goal hypotheses ruled out, in this case *permanently*. Finally, we investigate the trajectories with the ego using the robust baseline in Figure 9. As the human strategically moves to the center of the road to maximize the chance of collision, the planning algorithm fails to find any safe passing "corridor" before times out, but controls the ego vehicle to stop, resulting in a "freezing robot" situation [50].

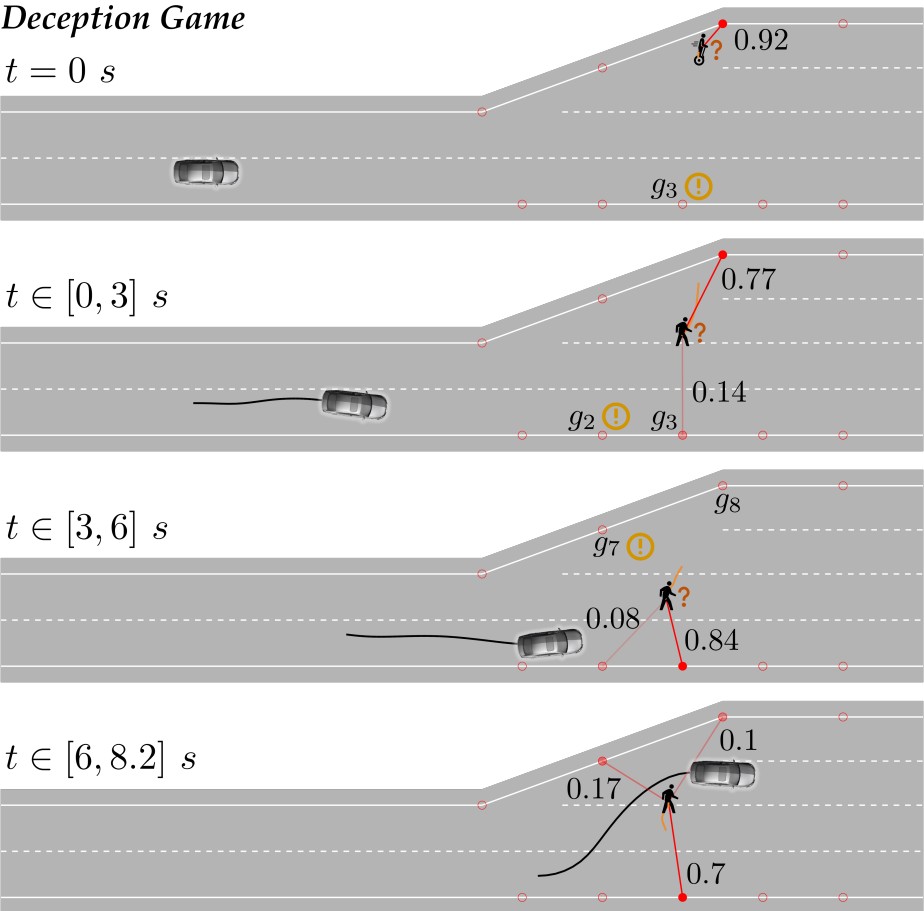

**Figure 6:** Simulation snapshots for Appendix B. Both the robot and the human use the Deception Game policy synthesized with adversarial RL. The human is allowed to take an action that violates the inference hypothesis (Assumption 1). Yellow exclamation marks indicate when the inference hypothesis is violated as the human takes an action corresponding to a discarded hypothesis. Red circles denote the human's goal hypotheses and their color intensity represents their belief probability. A line drawn between the human and a goal hypothesis $g_i$ means that $b(g_i) \geq \epsilon_\theta = 0.05$. The numbers denote the probabilities of hypothesized goals. The pedestrian and Segway icons indicate the ground-truth human type. The orange question mark implies that neither hypothesis has a probability below the threshold. The ego vehicle managed to safely complete the task even if the opponent human was not conformant to the inference hypothesis and the prior belief was biased towards the human not crossing the road.

## C  Deception Game with Motion Transformer

### C.1  Problem Setup

Consider the traffic scenario when the robot aims to traverse the intersection without violating the road bound or causing a collision with the adversary. We model both vehicles using the kinematic bicycle dynamics with longitudinal acceleration and steering angle controls. In addition, we limit their state and action space by bounding velocity $v \in [0, 10]$ m/s, acceleration $a \in [-5, 5]$ m/s$^2$, and steering angle $\delta \in [-0.5, 0.5]$ rad.

To infer the adversary's future actions, the robot uses a state-of-the-art trajectory prediction model, Motion Transformer (MTR) [1], which outputs a Gaussian Mixture Model of trajectories over the next 8 seconds from the 1.1 seconds of scene history. To construct the inferred control bound, we utilize a proportional controller to track the mean trajectory of each mixture component (mode $\theta$) as the adversary's nominal policy $\pi_t^o(x_t; \theta)$. In addition, we set $d_t^o(\theta)$ by assuming it can deviate from the nominal policy up to $\pm 2$ m/s$^2$ in acceleration and $\pm 0.1$ rad in steering angle. Since MTR outputs

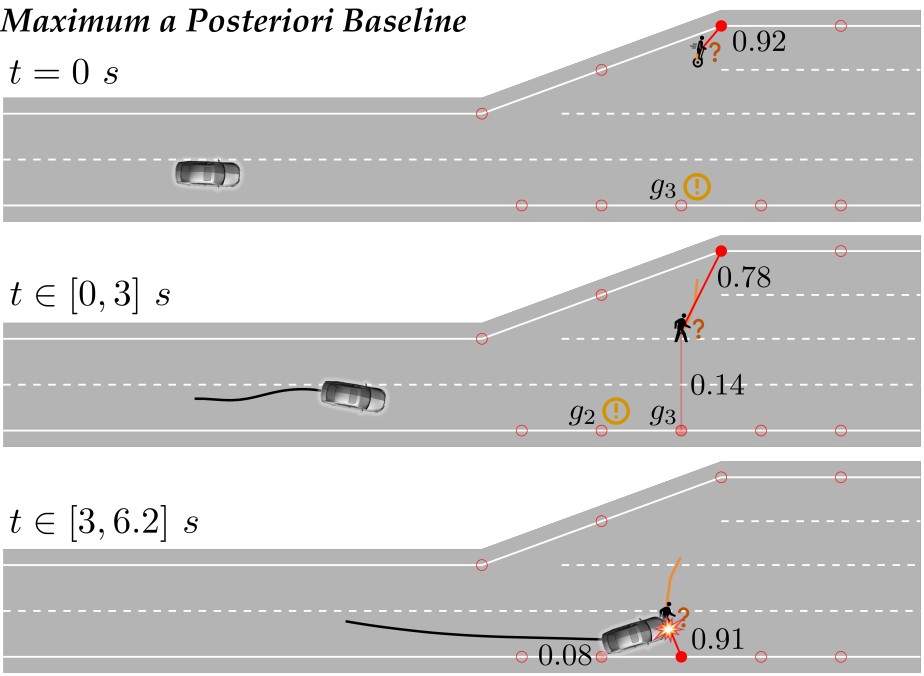

**Figure 7:** Simulation snapshots for [Appendix B](#) with the ego vehicle applying the MAP baseline policy. The human uses the Deception Game policy synthesized with adversarial RL, and is allowed to take an action that violates the inference hypothesis ([Assumption 1](#)). The ego vehicle collided with the human at $t = 6.2$ s.

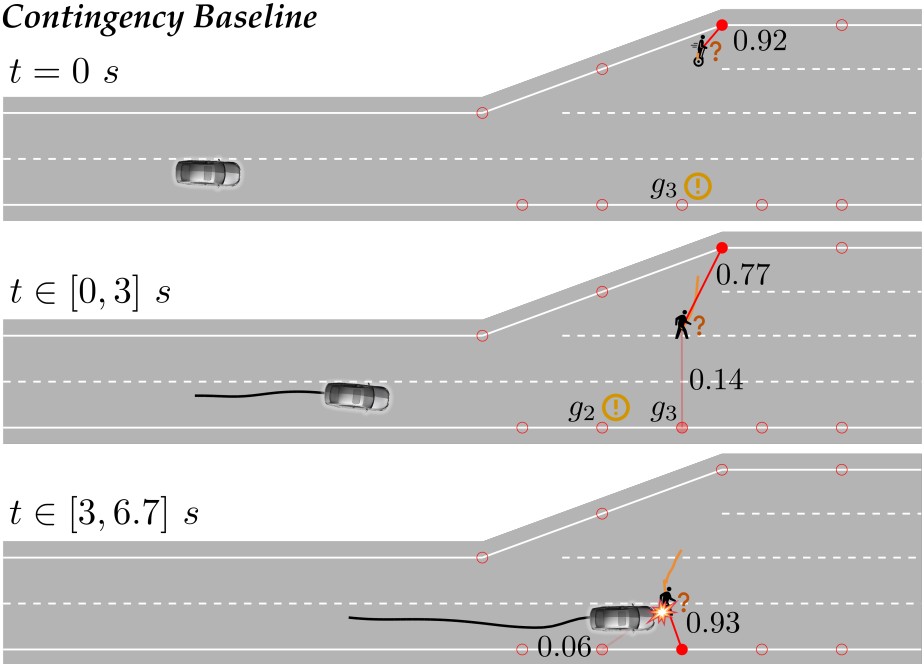

**Figure 8:** Simulation snapshots for [Appendix B](#) with the ego vehicle applying the contingency baseline policy. The human uses the Deception Game policy synthesized with adversarial RL, and is allowed to take an action that violates the inference hypothesis ([Assumption 1](#)). The ego vehicle collided with the human at $t = 6.7$ s.

64 modes using a prior motion query, we aggregate overlapping trajectories using non-maximum suppression and mask out modes with $b_t(\theta) < 0.05$.

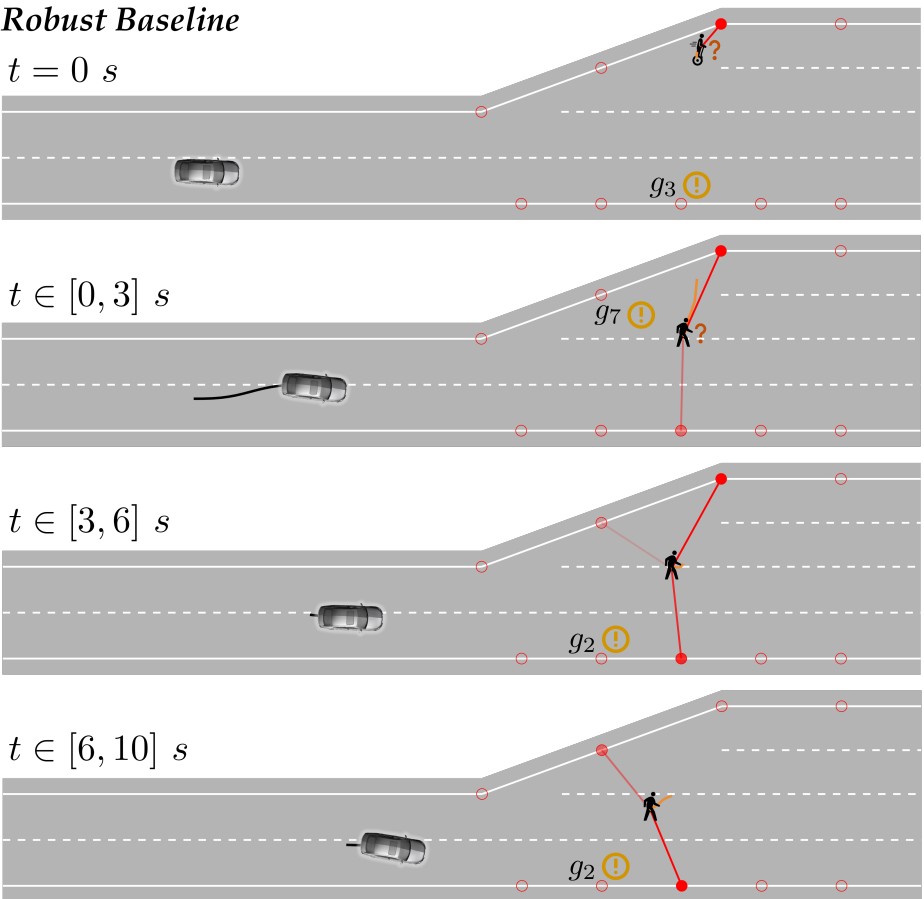

**Figure 9:** Simulation snapshots for [Appendix B](#) with the ego vehicle applying the robust baseline policy. The human uses the Deception Game policy synthesized with adversarial RL, and is allowed to take an action that violates the inference hypothesis ([Assumption 1](#)). The ego vehicle failed to pass the road section in 10 s.

## C.2 Network Architecture and Training Stratergy

The MTR model is first trained with the entire Waymo Open Motion Dataset [42] for 30 epochs and achieves claimed results in their paper. Then, we set up a simulation environment with the pre-trained MTR model in the loop to generate predictions and inferred control bounds for the adversaries. We represent the state with the absolute pose of the robot w.r.t the map, the adversary's relative pose w.r.t the robot, the nominal actions for each valid prediction mode, and their probabilities. The Deception Game is trained using the Iterative Soft Adversarial Soft Actor-Critic (ISAACS) framework, where four neural networks are trained asynchronously.

- The *ego actor* is the policy of the robot. It first encodes the states and each prediction mode independently by multi-layer perceptions (MLP). Then the state feature is concatenated to each prediction feature and passed through another MLP. We conduct max-pooling across all prediction modes to generate the aggregated feature, which is processed by the final MLP and becomes the mean and standard deviation of the robot's action. Finally, we sample the action using the squashed Gaussian distribution described in Soft Actor-Critic [51].

- The *adversarial actor* is the policy of the adversary. It first encodes the states, the action of the robot, and each prediction mode independently by MLPs. Then we concatenate the state, action, and each prediction features, which are later aggregated by another MLP. The final MLP processes the aggregated feature and outputs each prediction mode's mean,

standard deviation, and probability. We sample the adversary's action using a mixture of the squashed Gaussian distribution to enforce the inferred control bound.

- The *static critic* is a simple MLP return the $Q$ value of the robot **only** considering the road boundary and target set.

- The *interaction critic* returns the *residual $Q$* value of the interaction between the robot and the adversary. It first encodes the states, the action of both actors, and each prediction mode independently by MLPs. Then we concatenate the state, action, and each prediction features, and generate the aggregated feature of each mode through an MLP. The final MLP processes this feature and outputs $Q$ values for each prediction mode.

Unlike the standard ISAACS procedure, the *ego actor* and *static critic* are first trained by ignoring the collision with the adversary. Then we train *ego actor*, *static critic*, and *adversarial actor* jointly through domain randomization by randomly sampling the initial states of both actors and the adversary's action from its inferred control bound. In this process, We take the largest $Q$ value from *adversarial actor* among all modes, activate it through the SoftPlus function, and add it to the output from the *static critic* as the final $Q$ value for the robot. In this way, the resulting $Q$ value is strictly equal to or larger than that from the *static critic* as the robot will lose the game regardless of the adversary's state when violating the road constraint. Through our experiment, we found that pre-training the *ego actor* and *static critic* is necessary to stabilize learning process.

### C.3   Additional Results

A close-up in Figure 10 confirms that the adversary using the Deception Game policy indeed took adversarial actions to *actively* shift the trajectory predictions to intercept the robot. This enables the adversary to pursue the robot within the predictive action bound and indicates that our framework can capture the implicit learning dynamics a deep neural network represents. Moreover, even with one of the state-of-the-art trajectory predictors, small deviations from the nominal action can dramatically shift the robot's belief in the adversary over a short period. Therefore, classical methods (including the baseline ILQR policy) that rely on "accurate" trajectory prediction in an open-loop fashion [52] and frequently replan are insufficient for safe interactions. The intelligent robot needs to close the loop of learning by accounting for its future ability to learn and adapt.

Due to road boundaries and maneuverability limitations, the robust policy could not find feasible actions, as shown in Figures 11 and 12. The ILQR baseline failed all tests as it cannot account for the adversary's ability to change the robot's prediction in the future. Since our Deception Game policy is trained for intersections with randomized initial states of both vehicles, it is generalized to safely navigate different scenarios.

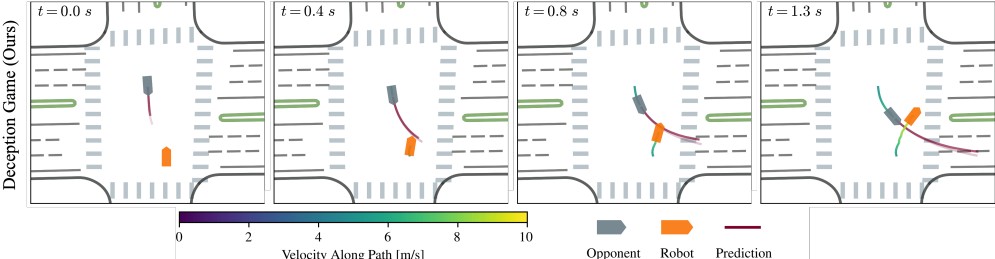

**Figure 10:** Case study: neural predictor (200D). Our framework can model an adversary whose policy adversarially influences the robot's implicit learning dynamics.

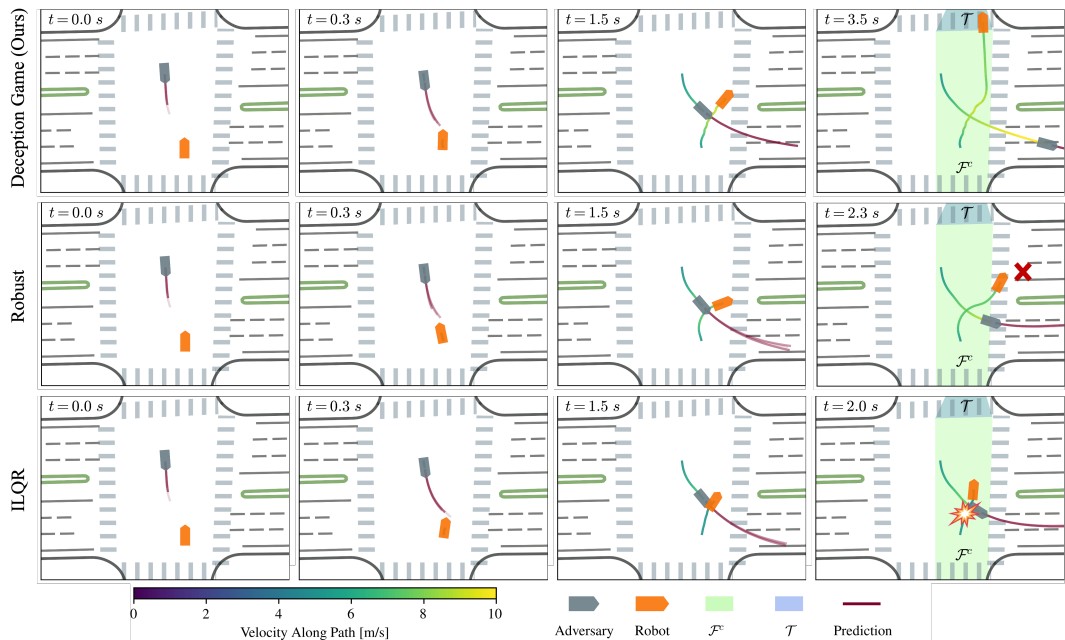

**Figure 11:** The adversary made an unprotected left turn when the oncoming robot entered the intersection. Robots using Deception Game (top) safely reached the target $\mathcal{T}$ by taking a proactive action even when the adversary was predicted to yield. The Robust Policy (middle) overreacted to the adversary's action, violated the road boundary constraints, and entered its failure set $\mathcal{F}$. The ILQR policy (bottom) was overly optimistic about the prediction and caused a collision

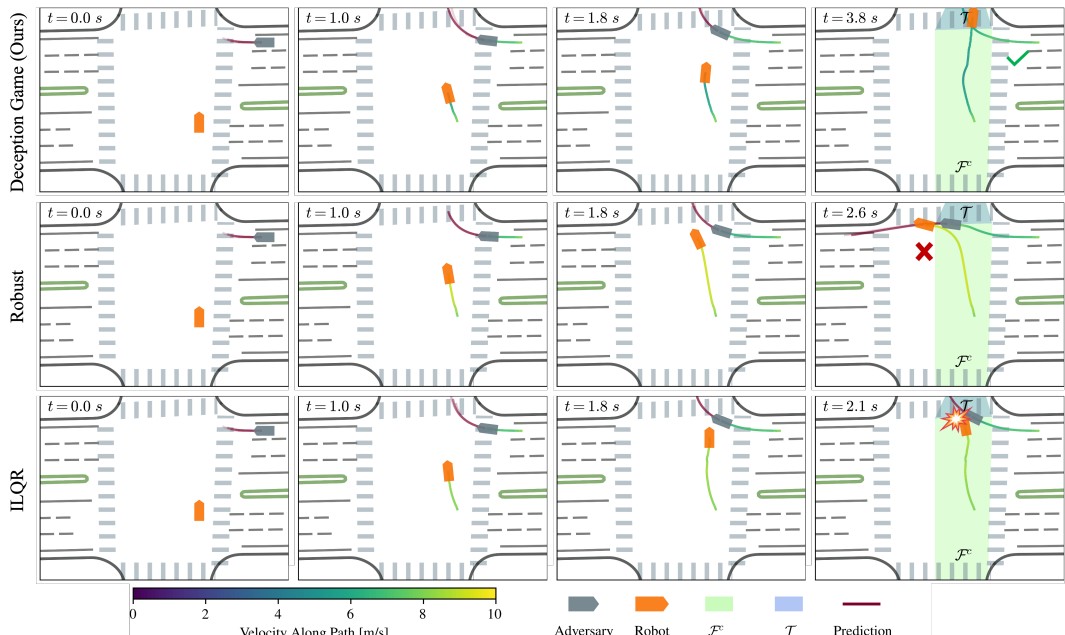

**Figure 12:** The robot interacted with the merging adversary. Robots using Deception Game (top) safely reached the target $\mathcal{T}$ by yielding to the adversary. The Robust Policy (middle) overreacted to the adversary's action, violated the road boundary constraints, and entered its failure set $\mathcal{F}$. The ILQR policy (bottom) was overly optimistic about the prediction and caused a collision

In Figure 13, we compared the robot's behaviors under different policies when interacting with a non-adversarial vehicle replaying the trajectory from Waymo Open Motion Dataset. We observed all three policies successfully traversed through the intersection. Both the proposed Deception Game and the robust policy showed similar evasive behaviors, where the robot using the robust policy accelerated much faster. The ILQR demonstrated closer maneuvers to the robot's ground truth trajectory from the dataset.

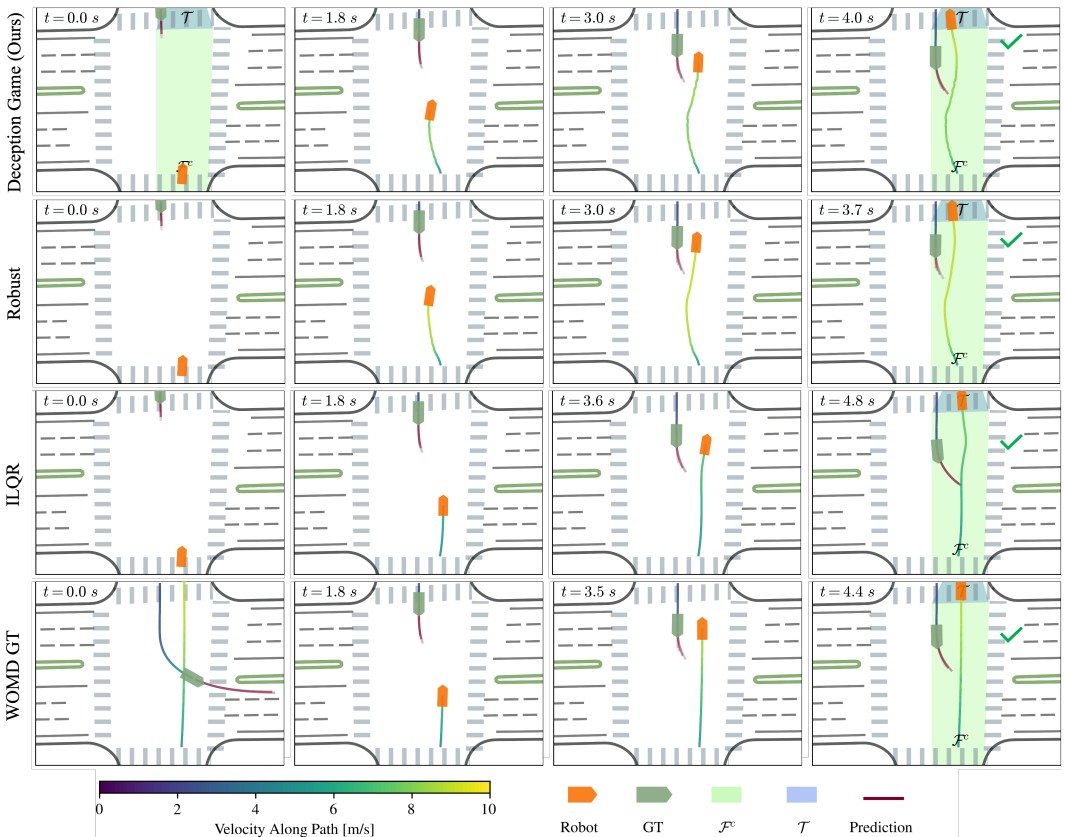

**Figure 13:** Robot's behaviors under different policies when interacting with a non-adversarial vehicle replaying the trajectory from Waymo Open Motion Dataset.

Under commonly used pairwise decomposition assumption [43, 44], Figure 14-17 demonstrate our proposed Deception Game's scalability to multiple agents scenarios. In these examples, We treated the closest vehicle to the ego as the adversary. The ego and the adversarial agent used policies from the Deception Game, while the third nonplaying vehicle applied the MTR nominal action.

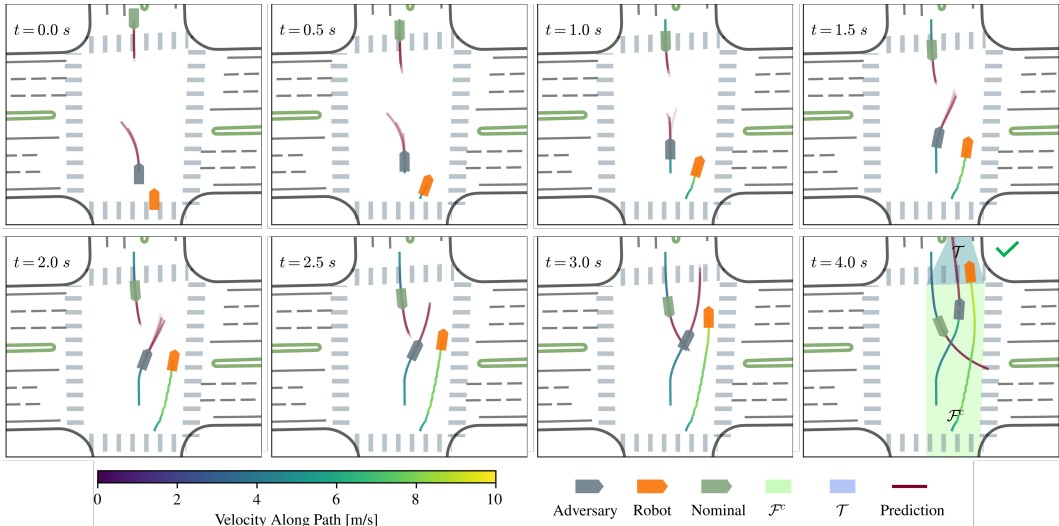

**Figure 14:** The robot interacted with the merging adversary while a nonplaying vehicle (NPC) attempted to turn left from the opposite lane. The nonplaying vehicle applies the most likely control from MTR at each timestep.

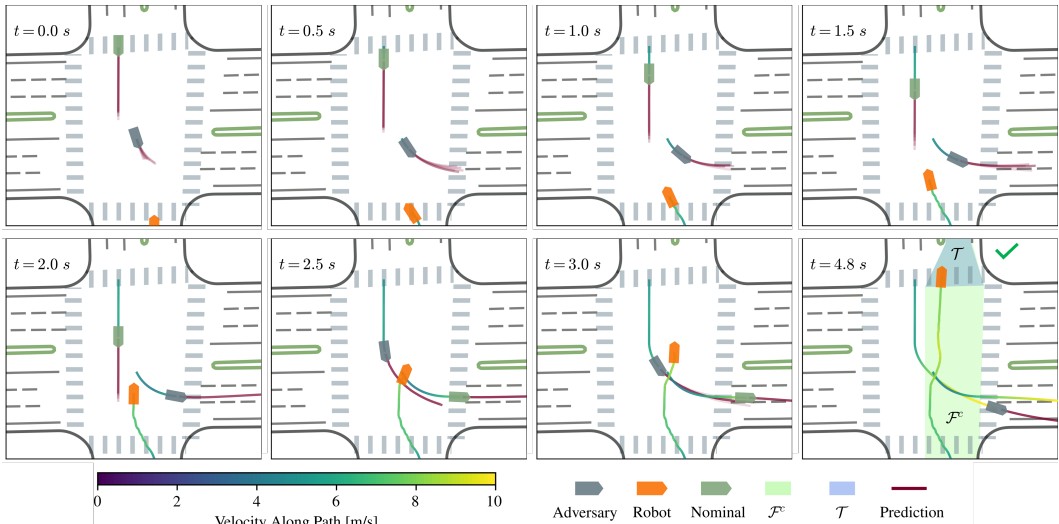

**Figure 15:** The robot interacted with two consecutive left-turning adversaries. Under the pair-wise assumption, we treated the closest vehicle to the robot as the adversary at each timestep. The further vehicle is a nonplaying character at each timestep and applies the most likely control from MTR.

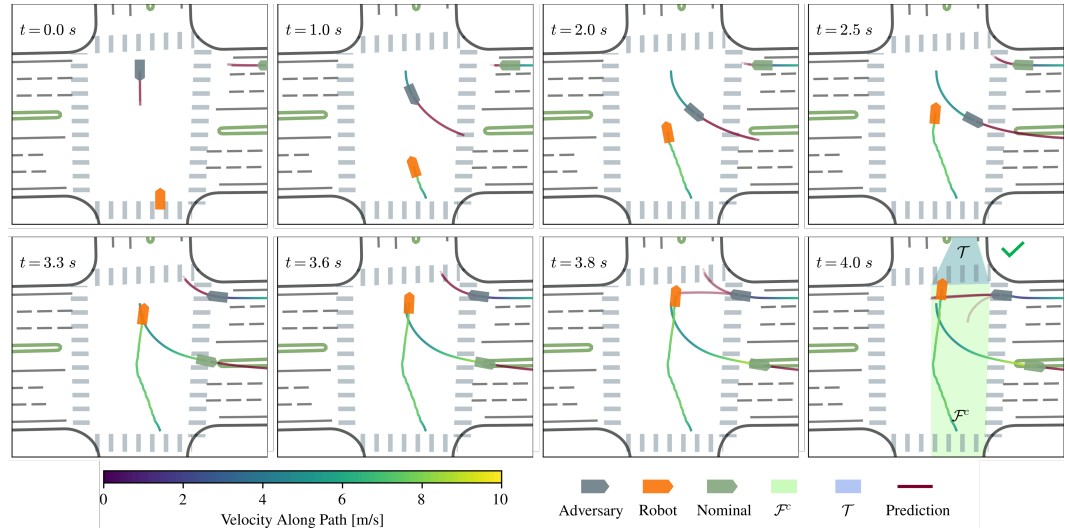

**Figure 16:** Under the pair-wise assumption, the robot interacted with the left-turning adversary by evading behind it. Then, the robot interacted with another merging adversary and reached the target without collision.

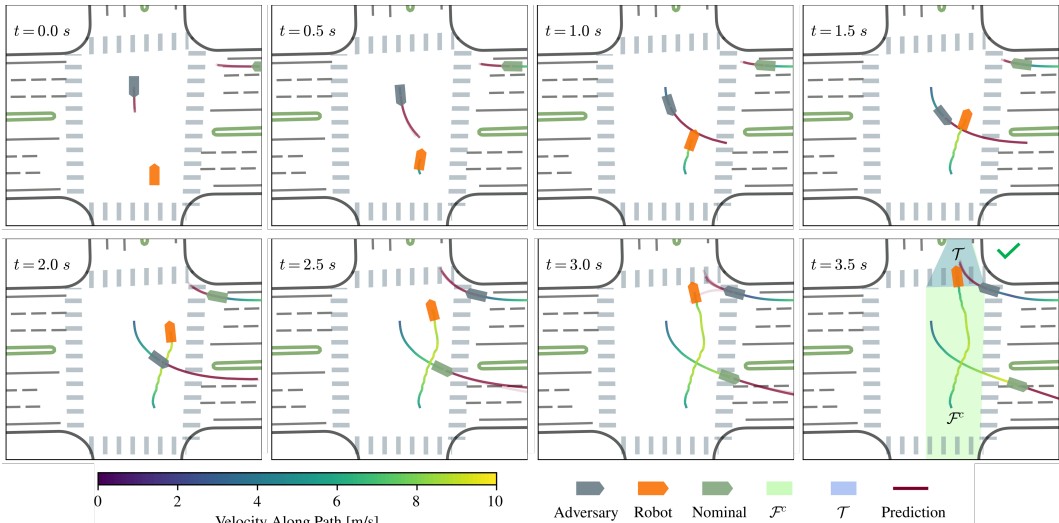

**Figure 17:** Under the pair-wise assumption, the robot interacted with the left-turning adversary by evading in front of it. Then, the robot interacted with another merging adversary and reached the target without collision.

