# OpenReview forum: "Deception Game: Closing the Safety-Learning Loop in Interactive Robot Autonomy"
_robot-learning.org/CoRL/2023/Conference — CoRL 2023 Poster_

### Official Review · Reviewer_BY31 · 2023-07-14

**Confidence:** 5
**Originality:** Good
**Technical Quality:** Very Good
**Clarity Of Presentation:** Good
**Impact:** 2

**Recommendation:**

Weak Accept: I recommend accepting the paper, but will not argue for my recommendation if the majority of other reviewers have a different opinion.

**Review:**

The approach targets the important problem of promoting safe behaviors of robots in environments with uncertain motions of other agents/humans.
The paper is generally well written, and the visualizations help readers to follow the approach.
The proposed approach is an interesting way to increase the safety by considering the current belief of how the environment evolves.
It is great to see that the authors formulate a hypothesis for their experiments and motivate why the design of the experiments helps to investigate the hypothesis.
The metrics are also explained well to readers.
The experiments are carried out in simulation in three different scenarios.
The attached video helps to understand the experiments and compare the approach with the chosen baselines.


The proposed approach has an interesting and relevant focus, but is not yet convincing.
The experimental validation needs to be vastly improved and the comparison to the existing safety literature needs more attention.
The paper misses to relate the approach to many existing papers on safe planning.
In fact, this paper focuses primarily on citing papers originating from Berkeley and with a similar touch to the proposed approach.
There are no mentions of other relevant safety works, e.g., done at Michigan and TUM (reachability), ETH and Caltech (safe control), or Stanford (risk-based approaches).
The paper mentions in L23 that existing safety approaches make static modeling assumptions.
However, many recent papers investigate this interactive component and dynamically ensure safety.
For instance, Brüdigam et al. (T-IV, 2023), Kwon et al. (HRI, 2020), Nishimura et al. (CoRL, 2022) or Lindemann et al. (RA-L, 2023), or reactive approaches such as REFINE from Liu et al. (arXiv, 2022).
The paper needs to better discuss the existing safety landscape.

One of the main motivations for the proposed approach is the focus on the current belief of how the situation will evolve.
The most likely predictions above the user-defined threshold are considered for planning.
The belief is modeled through a discrete set of hypotheses.
From the paper, it is not yet clear how such discrete sets can look like in typical robot applications.
The type of an agent or its goal are interesting, but it is not clear how one would choose and model this set.
For instance, how does one know which goals are important?
The use of the prediction in the AV examples also only cover a set of possible predictions.
What if those are incorrect at some point?
The approach adapts by recomputing the belief and updating the hypothesis.
But since less likely corner events are first discarded and not considered in the planner, the plan might navigate the robot into a dead-end in which it cannot react when such less likely behavior suddenly becomes likely.
This situation is not discussed in the paper.
For the central Proposition 1, there is no proof given in the paper or supplementary.
The computational complexity seems to scale with the number of hypotheses and the considered dynamical model.
What is the computational complexity of the approach?

The experiments somehow indicate that the approach is computationally heavy.
The first experiment uses simplified models and small set of hypotheses.
The scenario is chosen so that the agent crosses the road and interferes with the autonomous car.
Such a scenario often causes longer completion times for baselines that consider all hypotheses to ensure safety.
It would be great if the experiments were conducted in a Monte Carlo fashion on randomized scenarios.
With this simple scenario, it is hard to validate that the proposed approach is generally better.
In the high-dimensional experiment, it is weird that the autonomous vehicle first steers to the left into the way of the other agent even though the autonomous vehicle intends to cross the intersection.
Why is this the case?
Moreover, the paper claims that the trajectories of the robust approach are less smooth.
Although, the robust baseline generates an evasive maneuver, the proposed approach produces trajectories that constantly alternate in control inputs, always causing shaky trajectories.
This result needs further investigation.
This experiment also requires more scenarios (including more agents) to validate the approach.
Lastly, another safety baseline, such as a reactive control barrier function approach, would make the experiments stronger.

The limitations section is good and reveals shortcomings of the approach.
Yet, it should be improved by mentioning the persistent feasibility problems of the discrete set of hypotheses.
Also, it is not clear whether the paper focus on "safety analysis", "control synthesis", or "game theoretic planning". These terms are used interchangeably in the paper.

Other minor things are:
- In Fig. 1, it is not yet clear what the illustrated polygons show?
- Fig. 4 is not self-explanatory for readers, e.g., the changes, current belief, or planned trajectory.
- What are "road constraints" in the second experiment?
- The running example in L163 is not helpful for readers to understand the previous paragraph
- The demonstrations with Fig. 6 are not well described in the paper.


**Quality Of The Limitations Section:**

Additional details required

**Questions For Rebuttal:**

* What is the computational complexity of the approach?
* How can one ensure persistent feasibility?
* How robust is the approach to hypotheses that suddenly became likely and are not safe with respect to the current dynamics of the robot?
* How does the approach compare to other safety approaches in the literature?
* How does the approach perform in randomized scenarios?
* How does the approach perform with multiple other agents?
* Why does the vehicle steer into the adjacent lane even though it intends to cross the intersection?

**Robotics Focus:**

Relevant but unlikely to deploy to hardware in near future

**Summary Of Paper:**

This paper presents an approach to promote safe interaction between the robot and other agents/humans in the environment.
The idea is following a safe control policy that considers the scenario's uncertainty in the belief space.
To this end, the paper proposes to use belief games that optimize a safe policy given the current belief of how the scenario evolves.

**Summary Of Recommendation:**

The paper addresses the important problem of safe planning in uncertain interactive scenarios.
The underlying idea of using belief spaces is interesting.
Yet, the experimental validation is too simple (few hand-picked scenarios, one other agent, simple models) and does not show how the proposed approach performs in general scenarios.
The comparison with baselines also requires more attention.
The paper does not discuss the current safety literature and how the approach compares to those works.

---

### Official Review · Reviewer_dWyx · 2023-07-20

**Confidence:** 4
**Originality:** Fair
**Technical Quality:** Good
**Clarity Of Presentation:** Very Good
**Impact:** 3

**Recommendation:**

Weak Accept: I recommend accepting the paper, but will not argue for my recommendation if the majority of other reviewers have a different opinion.

**Review:**

Overall, the paper is well written. The problem
of safe control is important and relevant. I like
the systematic evaluation and explanation of each
benchmark.

The first comment I have is that no evaluation is
provided in a high-dimensional observation setting
where the proposed approach is presumably envisioned
to run. In particular, how are the evaluation
scenarios modeled exactly? Is there a simulator?
What is the object recognition accuracy? Also,
how does the number of objects affect the
algorithm's performance?

A second comment I have is that the RL framework
may result in quite unsafe controllers, as compared
to the more computationally heavy dynamic programming
one. The authors are encouraged to comment on
why and when the RL framework is not so useful in
terms of safety. Also, how different is the
computation time between the DP and the RL methods?
Judging from Table 1 and Figure 3, the difference
doesn't seem to be that big. Perhaps a more
involved scenario would be more convincing, where
the DP method fails to terminate within some
desired time (e.g., for run-time operation).

UPDATE AFTER REBUTTAL: The authors have addressed some of my concerns. Of course, the question of the ensuring the safety of RL controllers remains open. At the same time, absolute safety guarantees may require unnecessarily conservative policies. I think the paper has sufficient theoretical and experimental results to merit acceptance.

**Quality Of The Limitations Section:**

Additional details required

**Questions For Rebuttal:**

1. It seems that the RL framework may result in quite unsafe controllers, as compared to the more computationally heavy dynamic programming one. The authors are encouraged to comment on why and when the RL framework is not so useful in terms of safety.

2. Also, how different is the computation time between the DP and the RL methods? Judging from Table 1 and Figure 3, the difference
doesn't seem to be that big. Perhaps a more involved scenario would be more convincing, where the DP method fails to terminate within some desired time (e.g., for run-time operation).

**Robotics Focus:**

Highly relevant to robotics but no hardware experiments

**Summary Of Paper:**

This paper addresses the problem of developing a
safe controller in the presence of unpredictable
agents. In particular, the authors note that
static approaches such as Hamilton Jacobi
Reachability tend to be too conservative since
they do not factor in information available
at run-time. Thus, the authors propose a
belief-based safety analysis framework that
couples the system's current belief about
surrounding agents with model-free reinforcement
learning (RL). Evaluation is provided on 3 driving
benchmarks.

**Summary Of Recommendation:**

I think the paper has some interesting ideas, particularly the idea of using RL to overcome some of the scalability challenges of standard Hamilton Jacobi methods. I am only recommending a weak accept (as opposed to a strong accept) because the evaluation doesn't show a massive improvement in scalability (at significant costs in safety), so it is unclear whether the issue is just figuring out the correct RL setup or if there is a fundamental limitation to the proposed method.

---

### Official Review · Reviewer_6ETr · 2023-07-21

**Confidence:** 3
**Originality:** Good
**Technical Quality:** Good
**Clarity Of Presentation:** Very Good
**Impact:** 3

**Recommendation:**

Weak Accept: I recommend accepting the paper, but will not argue for my recommendation if the majority of other reviewers have a different opinion.

**Review:**

Overall, this paper tackles the challenging problem of robust, learning-aware planning under uncertainty: aiming to simultaneously be robust to uncertainty over latent parameters, while also accounting for how uncertainty may be reduced during interactions. The proposed approach is interesting, and well motivated and clearly explained, though fairly similar to the framework proposed by [10]. Empirically, the experiments demonstrate that the approach indeed reduces over-conservatism of a fully robust approach, and matches the performance of a MAP estimate based approach while simultaneously assuring safety. I also appreciated the discussion of how this approach could be scaled by using deep adversarial RL, and the demonstration that the approach did not need to rely explicitly on exact Bayesian inference, but could also work with the implicit approximate inference performed by state-of-the-art trajectory predictors which consume past agent trajectories.

One concern with this work is the scalability of the approach. Even with the proposed approximation techniques, the examples considered only considered a single non-ego agent with a limited set of latent parameter values. Furthermore, the approach yields a policy designed for a particular reach-avoid problem tailored to a particular scenario, and not a general policy. I would imagine that solving the adversarial RL problem is too computationally expensive to run online as a planner, and I'm not sure what value running this optimization on a single scenario provides in terms of "safety analysis." The paper would be strengthened with a discussion on how the results from the proposed approach could be used, similar to Bajcsy et al. [10].
Despite this, however, I think the problem formulation is interesting, and would be of interest to the robot learning community even if the algorithm cannot scale as is.

However, my main concern with this work is in the interaction between the adversarial assumption on other agents and the assumed belief updating dynamics.
- First, the authors show that the resulting ego policy learns to actively reduce uncertainty. However, the behavior this results in seems it may be overfit to adversarial agents. For example, to resolve ambiguity between a pedestrian and segway in Fig the agent chose to turn left, forcing an adversarial agent to move upwards toward the vehicle, something that only a pedestrian agent could do. However, in the real world, a pedestrian is unlikely to take such behavior -- in this way, it appears that the learned policy is taking information-gathering actions that work only for adversarial agents. I'm not sure if this behavior is desirable.
- Next, I'm curious about the implications of the belief threshold epsilon on the predictive control bounds. It would seem that choosing epsilon to be too large could break guarantees, as a true adversarial agent could take actions that the ego's Belief Game didn't account for. Are there any theoretical guarantees on safety that can be provided for the safety of the optimal policy obtained from the Belief Game? What are the implications of epsilon on these guarantees?



Minor comments:
- In line 144, it states that the reach-avoid set is visualized in Fig 1a. In line 81, it states the reach-avoid set is the set from which the ego agent is guaranteed a safe control strategy. However, in Fig 1a, the visualized sets are largest for the conservative approach, and smallest for the optimistic. but this feels backwards -- are the sets that are visualized actually the complements of the reach-avoid sets? i.e. the sets where the the assumptions predict a possible collision?

**Quality Of The Limitations Section:**

Limitations are addressed clearly

**Questions For Rebuttal:**

It would be interesting to see to what degree the resulting policies match real-world driving. Perhaps you could include results showing ego behavior in the traffic intersection scenario simulated with the other agent following its actual, real-world trajectory, and comparing the observed behavior of each of the policies against the real-world behavior of the ego?

How does performance depend on the accuracy of the belief updating model?

**Robotics Focus:**

Relevant but unlikely to deploy to hardware in near future

**Summary Of Paper:**

This paper presents an approach to considering safety in the presence of uncertainty over the dynamics of other agents in the scene. Rather than take a fully robust approach, the authors propose a method to account for the ego's ability to infer the latent parameters from future observations. Specifically, they propose a HJI reachability based approach which operates in the belief state dynamics. Their approach, Belief Game, formulates a reach avoid game in belief-state space, where the ego chooses actions to reach a goal while avoiding the other agents, while the other agents choose actions adversarially, constrained to a belief-state dependent control set. To address the high dimensional state space induced by accounting for the belief dynamics, they propose using adversarial RL to approximately solve the HJI equation. They demonstrate that this approach is less conservative than a fully robust approach while still ensuring safety, unlike MAP estimate-based planning approaches.

**Summary Of Recommendation:**

Overall, I think this paper presents an interesting approach to learning-aware safe planning. While the approach is presented clearly, and the experiments demonstrate that the learned policies can solve relatively simple interaction scenarios better than baselines, it's not clear to me that the resulting policies lead to realistic behavior, and I am still left with questions on how the results are meant to be used to support safety analysis.

The discussion period has been fruitful, and I appreciate the improvements the authors have made to their paper. I continue to support the acceptance of this work, and keep my score as is.

---

### Author Response · Authors · 2023-08-11
**Author Response Summary**

We would like to thank all the reviewers for their helpful comments and suggestions. We are delighted that all reviewers found our idea of learning-aware safety compelling and also, as noted by one reviewer, “appreciated the discussion of how this approach could be scaled by using deep adversarial RL, and the demonstration that [it] could also work with the implicit approximate inference performed by state-of-the-art trajectory predictors which consume past agent trajectories.”

We appreciate this opportunity to address some questions and make improvements to the manuscript (shown in blue in the revised version). Specifically, we made the following changes:

1. We clarified our inferred human control bound and the related Inference Hypothesis underpinning our formulation and differentiated our proposed method from scenario pruning approaches. We also added a concrete numerical example highlighting this difference in Appendix B.
2. We clarified the theoretical vs. practical scalability of our approach and emphasized how our case studies in Section 4 demonstrate our approach on 5-, 18-, and 200-dimensional joint physical-belief state spaces.
3. We clarified the relationship between “safety analysis,” “safe planning,” and “game theory” in our method. Specifically, in Section 3, we differentiated between the new safety analysis formulation, which is rooted in dynamic game theory, and the ability of our framework to synthesize safe robot policies. We also updated the visual framework schematic in Figure 2 for further clarification.
4. We ran additional experiments to shed light on how the resulting robot policies interact with multiple other vehicles, as well as real-world driving by replaying human driving data from the Waymo Open Motion dataset. These are in Appendix C.3.

Please note that throughout our responses we are using the citation numbers corresponding to references in the *updated* manuscript.

We look forward to continued discussion with the reviewers.

---

### Decision · Program_Chairs · 2023-08-30

**Decision:**

Accept (Poster)

**Comment:**

The paper introduces a novel way of synthesizing safe closed-loop control policies for robotic systems under uncertain future scenarios, using the framework of belief games. Reviewers agreed that the approach is interesting and novel, and valued the demonstrated results, especially after additional experiments and clarifications of the authors in their thorough rebuttal.